# Scalable Bayesian Active Learning with Batch Acquisition under Distribution Shift

## Abstract

The performance of machine learning models may suffer from significant decline when evaluated on the data exhibiting distribution shift. Although extensive research on algorithm design have been proposed, the acquisition of new data points to enlarge training datasets has also been verified as a promising solution path. Starting from this idea, we built our research upon bayesian active learning and propose a method that can efficiently acquire samples from a candidate pool of diverse data sources for improving performance on the shifted target population. Specifically, our method designs a novel acquisition function characterizing a Lower Bound of Batch Information Gain (LB-BatchIG) for target distribution and formulates batch acquisition as a submodular optimization problem. By resolving it with a greedy algorithm, we can determine the data batch from the candidate pool for annotation and training. Empirical studies on synthetic datasets and real-world datasets, including tabular data and image data, demonstrate that our batch acquisition algorithm can contribute to greater performance improvement than other algorithms.

## 1 Introduction

With the remarkable advancement, machine learning has been widely applied in many scenarios and achieved promising performance. Owing to the prominent capability, the models can minimize the predictive loss and achieve impressive performance on the data population of the same distribution to training data. Unfortunately, when applied in the wild applications, machine learning models can encounter the test data from a shifted distribution, which violates independent and identically distributed (i.i.d) assumption and induces notably performance deterioration.

To improve the generalization ability towards out-of-distribution (OOD) data, a bunch of researches have been proposed and offer promising solutions by more provable and carefully designed learning models, such as invariant learning (Arjovsky et al., 2019) and distributionally robust optimization (DRO) (Mohajerin Esfahani & Kuhn, 2018; Duchi & Namkoong, 2021), while maintaining the original training data. In contrast, from a data-centered perspective, recent researches (Liu et al., 2024; Fu et al., 2021) uncovered the great significance of collecting more samples for generalization. Motivated by this, we explore methods to augment the training data with additional samples, thereby improving generalization to *the shifted target distribution*. Given the labor costs and ethical concerns associated with data annotation, it is important to investigate how to achieve optimal generalization performance under a limited acquisition budget.

To this end, active learning offers a promising approach by querying samples that are beneficial to model performance from a candidate pool. More specifically, the candidate pool is expected to comprise diverse samples whose contribution to model performance varies. Within the acquisition budget, conventional active learning algorithms attempt to select the most informative samples based on acquisition criteria from distinct categories, such as uncertainty (Ducoffe & Precioso, 2018; Joshi et al., 2009; Ržička et al., 2020), model influence (Fukumizu, 2000; Zhang & Oles, 2000; Ash et al., 2020; Gal et al., 2017), and representativeness (Liu et al., 2016; Sener & Savarese, 2017b; Qin et al., 2021; Chattopadhyay et al., 2013; Yu et al., 2006; Yang et al., 2017). However, they are primarily designed without accounting for the distributional shift from training to target data. To bridge this gap, recent efforts in Active Domain Adaptation (ADA) (Rangwani et al., 2021; Prabhu et al., 2021) acquire samples *directly from the target domain*. This approach, nevertheless, imposes

another restrictive requirements on the candidate pool, serving as a condition that may not hold in many practical scenarios.

To overcome the above issues, we aim to *propose a scalable and provable* criteria towards selecting a batch of samples for generalization to shifted target distribution. Fortunately, we observe that the development of Bayesian active learning (Sun et al., 2015; Haut et al., 2018; Kirsch et al., 2019; Gal et al., 2017; Houlsby et al., 2011) offers new opportunities to guide our expected sample acquisition. As a representative, a acquisition criteria, termed EPIG (Smith et al., 2023), is proposed to measure the predictive information gain on target data distribution brought by a single, unlabeled sample. However, such criteria suffers from the severe dilemma on the **scalability issue**, as it only *supports selecting sample with the highest criteria*. Specifically, directly extending it by selecting the top K ones with the highest EPIG (notated EPIG) or stochastically selecting samples with EPIG-based probability (notated PowerEPIG) will tend to select similar samples and suffer from redundant information problem. As shown in Figure 1, these straightforward extensions can induce prohibitively *high mutual predictive information gain* among the samples in selected batch, especially directly selecting top K ones.

Therefore, in this paper, we propose to facilitate a more scalable Bayesian active learning criteria at batch-level. Specifically, our approach theoretically characterizes a Lower Bound of Batch Information Gain(LB-BatchIG) for target population, thus avoiding the trivial sum of individual criteria on samples (Smith et al., 2023). As the number of potential batches is **combinatorial** relative to the pool size and batch size, identifying the optimal batch for our LB-BatchIG exhibits high time complexity if one enumerates the whole the candidate batches directly. To further improve the algorithm efficiency, we theoretically prove the submodular property of this combinational optimization problem. Consequently, an efficient greedy solution can be derived with the time complexity polynomial to the size of data pool and batch. Finally, the acquired batch by our LB-BatchIG can be guaranteed at least $(1 - \frac{1}{e})$-optimal ($e$ is the natural constant) to the optimal (oracle) batch.

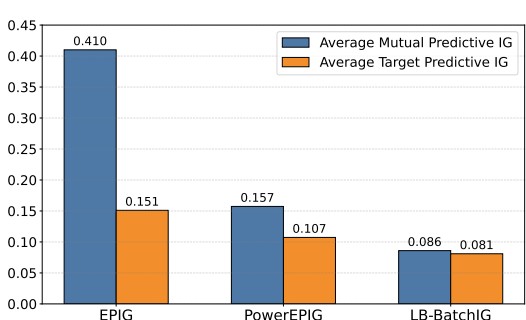

Figure 1: The results are obtained on the synthetic experiments (see details in section 4) with $\alpha = 1200$ and acquisition size $c = 50$. Mutual predictive IG refers to the predictive information gain provided to other samples within the batch, while target predictive IG denotes the individual information gain contributed to the target data distribution.

We conduct extensive experiments on synthetic datasets and real-world datasets, including tabular data and image data. In each experiment, the test dataset adheres to the target distribution distinct from both the training dataset and candidate pool. We compare our algorithm with baselines by iteratively acquiring new samples of given budget for annotation, adding them to the training data, and retraining the model. The experimental results show that with the same acquisition budget, the model trained on the samples selected by our algorithm achieves superior performance than other methods. The main contributions of this paper can be summarized as following:

- To the best of our knowledge, we are the first to investigate tractable acquisition function at batch-level for enhancing the model performance under distribution shift.

- We propose a novel acquisition function denoted as LB-BatchIG which supports acquiring data batch efficiently. We prove this function satisfies the submodular property and propose a greedy algorithm to maximize it.

- We conduct extensive experiments on synthetic datasets and real-world datasets to verify the effectiveness of our proposed algorithm.

## 2 PRELIMINARIES

In this section, we introduce our problem formulation and bayesian active learning which is the cornerstone of our method.

## 2.1 PROBLEM FORMULATION

We denote $\mathbf{X} \in \mathbb{R}^d$ as covariates and $\mathbf{y} \in \mathcal{Y}$ as label. This paper focuses on classification problem, thereby $\mathcal{Y} = [K]$. The initial training dataset is given as $\mathcal{D}_{tr} = \{(\mathbf{x}_i^{tr}, y_i^{tr})\}_{1 \leq i \leq n_{tr}}$. To enhance the model performance on the shifted target distribution $p^t(\mathbf{X}, \mathbf{y})$, we aim to select samples from a large unlabeled pool $\mathcal{D}_{po} = \{\mathbf{x}_i^{po}\}_{1 \leq i \leq n_{po}}$, annotate them and supplement them into training dataset. The acquisition process consists of $t$ rounds, and in each round $c$ samples are acquired. Therefore, the total acquisition budget is $g = t \cdot c$. The closed-form probability density of test distribution $p^t(\mathbf{X})$ is usually difficult to obtain. Hence, we denote the target distribution as the collection of test samples $\mathcal{D}_{te} = \{\mathbf{x}_i^{te}\}_{1 \leq i \leq n_{te}}$.

## 2.2 BAYESIAN ACTIVE LEARNING

Bayesian active learning borrows the idea of bayesian experimental design (Lindley, 1956; Chaloner & Verdinelli, 1995) which quantifies the information gain of interest variables from experiments.

To adapt bayesian experimental design to active learning, the designed experiment is defined as covariate $\mathbf{X}$ and the outcome of experiment is defined as the label $\mathbf{y}$. Assuming the predictive model $f_\theta(\mathbf{y}|\mathbf{X})$ with parameter $\theta$ characterizes the dependency between $\mathbf{X}$ and $\mathbf{y}$, BALD (Houlsby et al., 2011; Gal et al., 2017; Kirsch et al., 2019) takes the model parameter $\theta$ as the interest variables and search the sample $\mathbf{x}$ that can reduce the entropy of $\theta$ to the maximal extent. Formally, it can be formulated as:

$$\arg \max_{\mathbf{x}} H[\theta|\mathcal{D}] - \mathbb{E}_{p(\mathbf{y}|\mathbf{x};\mathcal{D})}[H[\theta|\mathcal{D}, \mathbf{x}, y]], \tag{1}$$

where $H(\cdot)$ is the shannon entropy, $\mathcal{D}$ is the dataset has been annotated and $p(\mathbf{y}|\mathbf{x};\mathcal{D}) = \mathbb{E}_{p(\theta|\mathcal{D})}[p(\mathbf{y}|\mathbf{x}, \theta)]$ is the posterior predictive distribution marginalized over model parameter $\theta$. However, (Smith et al., 2023) pointed out that predictive uncertainty is mismatched with parameter uncertainty, and therefore propose an acquisition function of prediction-oriented manner. The acquisition function expected predictive information gain (EPIG) is designed for single data point.

$$\text{EPIG}(\mathbf{x}) = \mathbb{E}_{p^t(\mathbf{x}^*), p(\mathbf{y}, \mathbf{y}^*|\mathbf{x}, \mathbf{x}^*; \mathcal{D})} \left[ \log \frac{p(\mathbf{y}, \mathbf{y}^*|\mathbf{x}, \mathbf{x}^*; \mathcal{D})}{p(\mathbf{y}|\mathbf{x}; \mathcal{D}) p(\mathbf{y}^*|\mathbf{x}^*; \mathcal{D})} \right], \tag{2}$$

where $p^t(\mathbf{x}^*)$ represents the covariate distribution of target population.

Repeatedly acquiring only one sample with highest EPIG score would result in excessively many acquisition rounds. Though we can directly adapt this method to select samples of top-K EPIG value and consequently reduce the acquisition rounds, it would suffer from redundant information problem and lead to sub-optimal performance.

Summarily, it is essential to propose a new acquisition function for batch samples that considers simultaneously the reduction of predictive uncertainty on target population as well as the diversity of batch.

## 3 PROPOSED METHOD

In this section, we firstly propose our acquisition function LB-BatchIG. Then we conduct theoretical analysis to reveal that it satisfies the sub-modular property. Based on this theoretical finding, we can apply greedy algorithm to resolve the problem of searching batch with maximum LB-BatchIG.

### 3.1 ACQUISITION FUNCTION: LB-BATCHIG

By extending the idea of bayesian active learning, we characterize the information gain brought by batch samples $\{\mathbf{x}_i^b\}_{1 \leq i \leq c}$ to the prediction on the target population $p^t(\mathbf{x}^*)$ as follows:

$$\mathbb{E}_{p^t(\mathbf{x}^*)} \left[ \text{BIG}(\{\mathbf{x}_i^b\}, \mathbf{x}^*) \right], \tag{3}$$

where the function $\text{BIG}(\cdot)$ is defined as:

$$\text{BIG}(\{\mathbf{x}_i^b\}, \mathbf{x}^*) = \mathbb{E}_{p(\mathbf{y}_1^b, \ldots, \mathbf{y}_K^b, \mathbf{y}^*|\mathbf{x}_1^b, \ldots, \mathbf{x}_K^b, \mathbf{x}^*; \mathcal{D})} \left[ \log \frac{p(\mathbf{y}_1^b, \ldots, \mathbf{y}_K^b, \mathbf{y}^*|\mathbf{x}_1^b, \ldots, \mathbf{x}_K^b, \mathbf{x}^*; \mathcal{D})}{p(\mathbf{y}_1^b, \ldots, \mathbf{y}_K^b|\mathbf{x}_1^b, \ldots, \mathbf{x}_K^b; \mathcal{D}) p(\mathbf{y}^*|\mathbf{x}^*; \mathcal{D})} \right]. \tag{4}$$

However, the numeric value of Equation 3 is difficult to exactly calculate since it involves the joint probability density estimation of multiple variables. Therefore, we propose an alternative measurement as the substitute, which bypasses the calculation of multi-dimensional probability and can be significantly easier to compute. Specifically, the formula of the substitute measurement is

$$\text{LB-BatchIG}(\{\mathbf{x}_i^b\}) = \mathbb{E}_{p^t(\mathbf{x}^*)}\left[\max_{1\le i\le c}\text{IG}(\mathbf{x}_i^b, \mathbf{x}^*)\right], \tag{5}$$

where the function $\text{IG}(\cdot)$ is

$$\text{IG}(\mathbf{x}, \mathbf{x}^*) = \mathbb{E}_{p(\mathbf{y},\mathbf{y}^*|\mathbf{x},\mathbf{x}^*;\mathcal{D})}\left[\log \frac{p(\mathbf{y},\mathbf{y}^*|\mathbf{x},\mathbf{x}^*;\mathcal{D})}{p(\mathbf{y}|\mathbf{x};\mathcal{D})p(\mathbf{y}^*|\mathbf{x}^*;\mathcal{D})}\right]. \tag{6}$$

It can be proved that the measurement $\text{LB-BatchIG}(\{\mathbf{x}_i^b\})$ is a lower bound of Equation 3. Hence, we can achieve the high predictive information gain brought by batch samples $\{\mathbf{x}_i^b\}$ through maximizing the objective of LB-BatchIG. Formally, it can be theoretically revealed with the following theorem.

**Theorem 3.1.** *According to the information theory, we have*

$$\text{LB-BatchIG}(\{\mathbf{x}_i^b\}) \le \mathbb{E}_{p^t(\mathbf{x}^*)}\left[\text{BIG}(\{\mathbf{x}_i^b\}, \mathbf{x}^*)\right]. \tag{7}$$

The detailed proof can be found in the section of appendix B.

## 3.2 SUB-MODULAR PROPERTY OF LB-BATCHIG

The optimization of LB-BatchIG is a combinatorial search problem with the number of potential solutions $\mathcal{C}_{n_{po}}^c$. The brute force method that directly enumerates the candidate batches and selects the one of the highest score is of exponential time complexity and computationally expensive. Therefore, it is in urgent need to design an efficient algorithm which produces a near-optimal solution with less computational cost.

The submodular property of the acquisition function brings the opportunity to solve the batch optimization problem with polynomial complexity. The definition of submodular function is as follows:

**Definition 3.1.** *Given a set $V = \{v_1, v_2, ..., v_m\}$, a function $f : 2^V \to \mathbb{R}$ taking subset of $V$ as input is a submodular function if the inequality holds:*

$$f(A) + f(B) \ge f(A \cap B) + f(A \cup B), \forall A, B \subset V \tag{8}$$

The literature (Nemhauser et al., 1978) unveils that the optimization of normalized monotone non-decreasing submodular function with cardinality constraint, formally $\max_{S \subset V, |S|=c} f(S)$, can be resolved by greedy algorithm. The resulting solution approximates the optimal one with a factor at least $1 - 1/e \approx 0.632$. In this way, the time cost is significantly reduced.

Fortunately, through theoretical analysis, we can prove the proposed acquisition function LB-BatchIG satisfies the submodularity property.

**Proposition 3.1.** *Regarding the unlabeled pool as the element set $V$ in Definition 3.1 and the sample batch as the subset, the acquisition function LB-BatchIG is a normalized monotone non-decreasing submodular function.*

The detailed proof can be found in the appendix B. Based on this promising property of our criterion, we design an efficient greedy algorithm to pursue a near-optimal sample batch.

## 3.3 IMPLEMENTATION

We successively introduce the details of our algorithms, including the acquisition function estimation and the batch construction process.

**LB-BatchIG Estimation** We firstly repeatedly draw a series of model parameters $\{\theta_l\}_{1\le l\le m}$ from posterior distribution and calculate two matrices $\mathbf{O} \in \mathbb{R}^{K \times K}$ and $\mathbf{Q} \in \mathbb{R}^{K \times K}$:

$$o_{i,j} = \frac{1}{m}\sum_{l=1}^m p(\mathbf{y}=i|\mathbf{x},\theta_l) \cdot p(\mathbf{y}=j|\mathbf{x}^*,\theta_l),$$

$$q_{i,j} = \frac{1}{m}\sum_{l=1}^m p(\mathbf{y}=i|\mathbf{x},\theta_l) \cdot \frac{1}{m}\sum_{l=1}^m p(\mathbf{y}=j|\mathbf{x}^*,\theta_l),$$

where $\theta_l \sim p(\theta|\mathcal{D}), 1 \leq l \leq m$. The values of $o_{i,j}$ and $q_{i,j}$ are the empirical approximations of $p(\mathbf{y} = i, \mathbf{y}^* = j|\mathbf{x}, \mathbf{x}^*; \mathcal{D})$ and $p(\mathbf{y} = i|\mathbf{x}; \mathcal{D}) \cdot p(\mathbf{y}^* = j|\mathbf{x}^*; \mathcal{D})$ respectively.

Based on the calculated matrices $\mathbf{O}$ and $\mathbf{Q}$, we can empirically estimate the function $\text{IG}(\mathbf{x}, \mathbf{x}^*)$ as follows:

$$\hat{\text{IG}}(\mathbf{x}, \mathbf{x}^*) = \sum_{i=1}^{K} \sum_{j=1}^{K} o_{i,j} \cdot \log \frac{o_{i,j}}{q_{i,j}}. \tag{9}$$

By sampling $\mathbf{x}^*$ from the target distribution $p^t(\mathbf{x}^*)$, we can estimate the acquisition function for $\{\mathbf{x}_i^b\}_{1 \leq i \leq c}$ through the equation:

$$\text{LB-BatchIG}(\{\mathbf{x}_i^b\}) \approx \frac{1}{s} \sum_{j=1}^{s} \max_{1 \leq i \leq c} \hat{\text{IG}}(\mathbf{x}_i^b, \mathbf{x}_j^*) \tag{10}$$

**Batch Construction** Owing to the promising property of normalized monotone non-decreasing submodular function, we propose a greedy algorithm to construct the acquisition sample batch. The algorithm consists of three steps.

- Firstly, the acquisition batch is initialized as an empty set. Formally, $\mathcal{B}_0 = \varnothing$.
- We iteratively add $c$ samples into the acquisition batch. In the $i^{th}$ iteration, we search for the sample $\mathbf{x}_{b_i}^{po}$ with the largest improvement of LB-BatchIG. Formally, this means

$$b_i = \arg \max_{1 \leq l \leq n_{po}} \text{LB-BatchIG}(\mathcal{B}_{i-1} \cup \mathbf{x}_l^{po}). \tag{11}$$

  Then the selected sample is incorporated in to the acquisition batch $\mathcal{B}_i = \mathcal{B}_{i-1} \cup \mathbf{x}_{b_i}^{po}$.

- Finally, the resulting batch $\mathcal{B}_c = \{\mathbf{x}_i^b\}_{1 \leq i \leq c}$ is the obtained batch for annotation.

The pseudo-code of the greedy algorithm can be found in the appendix D. After the batch $\mathcal{B}_c$ is obtained, the samples are removed from the unlabeled pool. The new pool is updated as $\mathcal{D}_{po} \leftarrow \mathcal{D}_{po} \setminus \mathcal{B}_c$. After annotating the oracle label $\{y_i^b\}_{1 \leq i \leq c}$ for $\mathcal{B}_c$, the samples are added into the training dataset, which means $\mathcal{D}_{tr} \leftarrow \mathcal{D}_{tr} \cup \{\mathbf{x}_i^b, y_i^b\}_{1 \leq i \leq c}$.

## 3.4 TIME COMPLEXITY

The time complexity of calculating function $\text{IG}(\mathbf{x}, \mathbf{x}^*)$ is $\mathcal{O}(mK^2)$. Computing the acquisition function for the batch of size $c$ requires $\mathcal{O}(cs)$ times of calculating IG function. Running our proposed algorithm consumes $\mathcal{O}(cn_{po})$ times of calculating LB-BatchIG. In contrast, the direct enumeration method needs to sweep all the $\mathcal{C}_{n_{po}}^c$ potential candidates. Since $c \ll n_{po}$, the times of LB-BatchIG calculation for enumeration method approximately equals $\mathcal{O}(n_{po}^c)$.

## 4 EXPERIMENT

We evaluate our proposed batch acquisition algorithm on diverse datasets, including synthetic data, tabular data and image data.

### 4.1 EXPERIMENTAL SETUP

**Baselines** To demonstrate the effectiveness of our proposed method, we implement the following baselines for comparision:

- *Uniform*: This method randomly selects $c$ samples from the candidate pool in each round without preference.
- *EPIG* (Smith et al., 2023): The original version of the method repeatedly select the sample with the highest expected predictive information gain (EPIG) score. To accommodate the batch acquisition setting, we simply adapt this method to rank the samples by EPIG score and choose the top-K samples in one round.

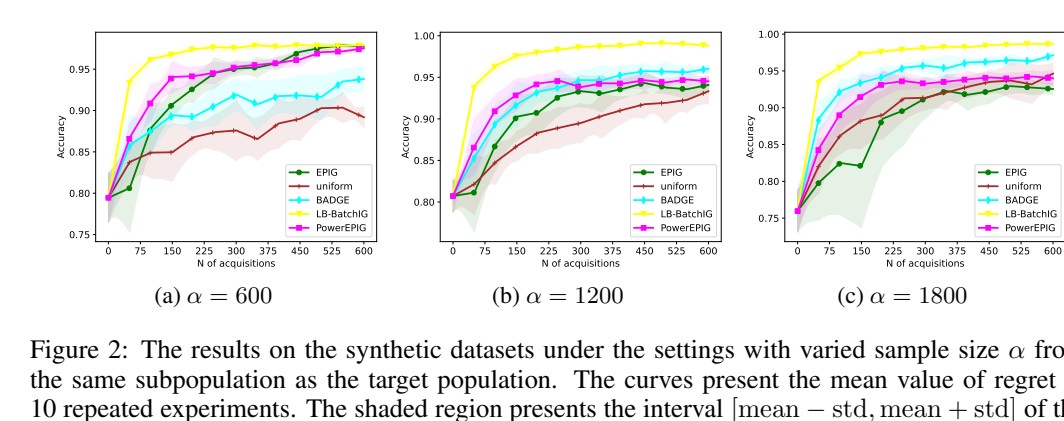

(a) $\alpha = 600$    (b) $\alpha = 1200$    (c) $\alpha = 1800$

Figure 2: The results on the synthetic datasets under the settings with varied sample size $\alpha$ from the same subpopulation as the target population. The curves present the mean value of regret in 10 repeated experiments. The shaded region presents the interval $[\text{mean} - \text{std}, \text{mean} + \text{std}]$ of the regret.

- *PowerEPIG* (Kirsch, 2021): It stochastically selects the samples with the normalized probabilities proportional to the EPIG score to the $\gamma^{th}$ power. Specifically, we set $\gamma = 5$ as the previous literature (Kirsch, 2021) suggested.

- *BADGE* (Ash et al., 2020): It computes the gradient-based embedding for each sample and runs k-means++ algorithm (Arthur & Vassilvitskii, 2007) to construct a batch of samples with diverse and representative embeddings.

**Evaluation metric** The data acquisition process consists of $t$ rounds. After each round, we retrain the predictive model on the enlarged training dataset including the original training samples and the samples selected by algorithms from the candidate pool. The accuracy of the retrained model on the target population for each round is recorded.

We repeat the above process several times and calculate the mean value and standard deviation of the accuracy across the repeated experiments.

**Model Setup** To enable uncertainty estimation, we adopt the MC Dropout (Gal & Ghahramani, 2016) technology for the predictive models. Specifically, the dropout layers are kept activated during the inference. Therefore, the random activation of dropout unit can be viewed as sampling from the posterior parameter distribution, and the prediction result can be varied across multiple inferences.

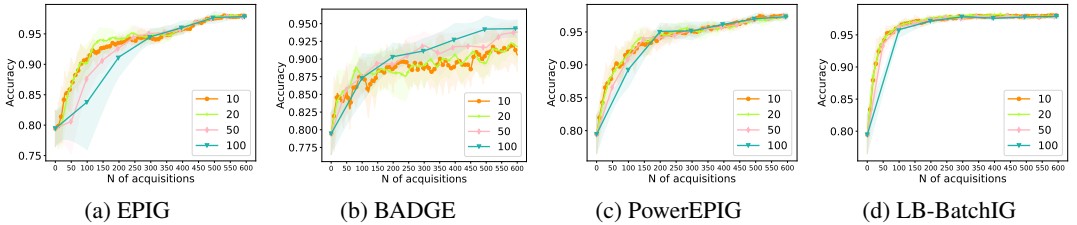

(a) EPIG    (b) BADGE    (c) PowerEPIG    (d) LB-BatchIG

Figure 3: The results on the synthetic datasets with varied batch size of acquisition. The experiments are conducted under the setting where $\alpha = 600$.

## 4.2 SYNTHETIC DATA

**Experimental setup** We generate the synthetic data of binary classification task. The samples come from three distinct data subpopulations. Specifically, the covariates $\mathbf{X} \in \mathbb{R}^d$ consists of two parts, the first $d_1 = d - 3$ elements of $\mathbf{X}$ are independently drawn from standard gaussian distribution:

$$x_{,1}, x_{,2}, ..., x_{,d_1} \overset{iid}{\sim} \mathcal{N}(0, 1).$$

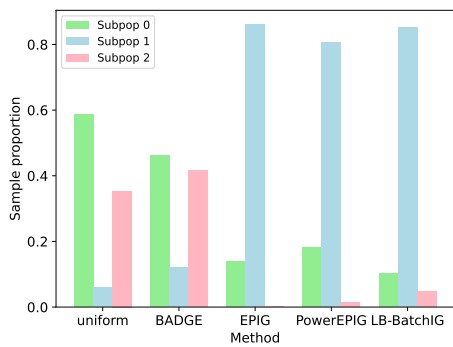

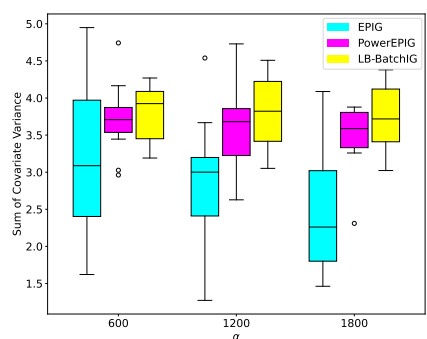

Figure 4: The average sample proportion from the subpopulations in an acquired batch. The results are calculated under the setting where $\alpha = 600, c = 50$.

Figure 5: The value distribution of batch diversity measurements across the acquisition process of EPIG, PowerEPIG and LB-BatchIG.

Each sample belongs to a subpopulation $s_i \in \{0, 1, 2\}$. Based on the covariates and subpopulation index, the ground truth labels $y_i \in \{0, 1\}$ are generated by the functions: $y_i = \mathbb{I}\left(\sum_{j=1}^{d_1} x_{i,j}\beta_{s_i,j} > 0\right)$, where $\mathbb{I}(\cdot)$ is the indicator function, and $\beta_k \in \mathbb{R}^{d_1}, 0 \leq k \leq 2$ are coefficient vectors specific to each subpopulation.

The element of $\beta_k$ is also drawn from standard gaussian distribution. The last three elements of covariates indicate the subpopulation index and represented as one-hot encoding. Specifically, we set $x_{i,d_1+s_i} = 1, x_{i,d_1+j} = 0, \forall j \neq s_i$.

We can induce distribution shift by adjusting the proportion of the subpopulations. Therefore, we construct the initial training datasets, candidate pools and target population with different compositions. The training datasets and target population are dominated by different subpopulations, while the candidate pool contains a large number of samples from different subpopulations. The detailed compositions are listed in the Appendix C.

For the model architecture, we adopt the neural networks consists of three fully connect layers with the hidden size equal to 32. We place dropout layer at the first hidden layer and set the dropout rate as 0.1.

**Results** We conduct repeated experiments for 10 times with varied sample size $\alpha$ of subpopulation $s = 2$. The sample acquisition process consists of $t = 12$ rounds, and in each round $c = 50$ samples are selected. The results can be found in Figure 2.

From the results, we find that uniform acquisition method achieves worst performance across the methods. This is because it neglects the distinction of samples in improving models and fails to identify the beneficial samples for training.

The EPIG and BADGE method can achieve better performance than Uniform since they consider the contribution difference to model among samples and try to select more beneficial samples. However, the improvement brought by them is limited because of the redundant information and distribution shift problem respectively. When the sample size $\alpha$ increases, the proportion of samples belonging to the same subpopulation as target in the candidate pool also increases. The distribution shift problem is less severe. Therefore, the performances of BADGE and Uniform both are improved. The PowerEPIG method introduces diversity property into EPIG, and mitigate the redundant information problem. However, it does not explicitly optimize the predictive information gain of a batch, and thereby achieve sub-optimal performance.

We display the average sample proportion from the subpopulations in an acquired batch in Figure 4. From the results, we can observe that EPIG, PowerEPIG and LB-BatchIG focus on the predictive information gain on the target population and successfully identify the samples belonging to the same subpopulation (i.e. subpopulation $s = 1$).

However, when the sample size $\alpha$ increases, the performance of EPIG degrades. This is mainly because that the larger $\alpha$ leads to larger sample density in the same covariate region, which facilitate

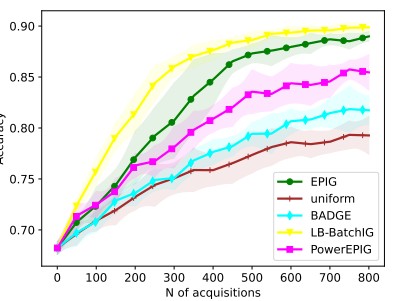
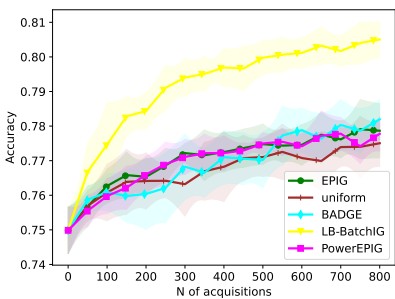

(a) Income Prediction, Target State PR     (b) Employment Prediction, Target State PR

Figure 6: The results on the tabular datasets under the different settings with different prediction tasks and state composition for training datasets, candidate pool and target population.

more severe redundant information problem. We aggregate the variance of the covariate elements $\{x_{,1}, x_{,2}, ..., x_{,d_1}\}$ of the acquired sample batch which measures the diversity of the selected samples. From the results in Figure 5, we observe that the diversity measurement of EPIG decreases with larger $\alpha$ which confirms our conjecture. In contrast, our proposed method achieves more diverse acquisition regardless of the composition of candidate pool, and thereby performs better than other methods.

**Batch Size Analysis** We change the batch size of single acquisition (as well as the acquisition rounds) to examine the effect of it on the performance curve. The results can be found in the Figure 3. EPIG and PowerEPIG underperforms with larger $c$ due to more severe redundant information problem, while our proposed method is robust to the batch size variation and achieves superior performance.

### 4.3 TABULAR DATA

Ding et al. (2021) construct a series of datasets from available US Census sources spanning over multiple years, states of the United States and various prediction tasks for the research on algorithmic fairness and distribution shift.

**Experimental setup** The data sources involve several prediction tasks. We choose the income and employment prediction as the benchmark to validate the effectiveness of methods. To create distribution shift, in this paper we leverage the available meta-information of states to constitute the different subpopulation composition. For each prediction task, we set up two experiment settings about different distribution shift respectively. The detailed information about the composition of the training dataset, candidate pool and target population can be found in the Appendix C.

We follow the same setup as the synthetic experiments and adopt the neural networks consists of three fully connect layers with the hidden size equal to 32. We place dropout layer at the first hidden layer and set the dropout rate as 0.1.

**Results** We repeat the experiments 10 times for each experimental setups about the prediction task and state compositions. The experimental results can be found in Figure 6. The overall trend is consistent with that of the synthetic dataset. On the whole, the Uniform performs worse than the other methods since it reflects no preference over the beneficial samples for performance improvement. Generally, the EPIG method achieves the second best performance especially in the income prediction task. However, in the employment prediction task, its advantage over other methods suffers from significant deterioration. It may be because in the employment prediction task, the samples with large predictive information gain tend to cluster and result in severe redundant information problem. In contrast, our method consistently accomplishes promising performance and outperforms the baselines.

### 4.4 IMAGE DATA

Domain generalization (Wang et al., 2022; Zhou et al., 2022) is an important branch of research developing the better performing models when encountering distribution shift. We leverage the

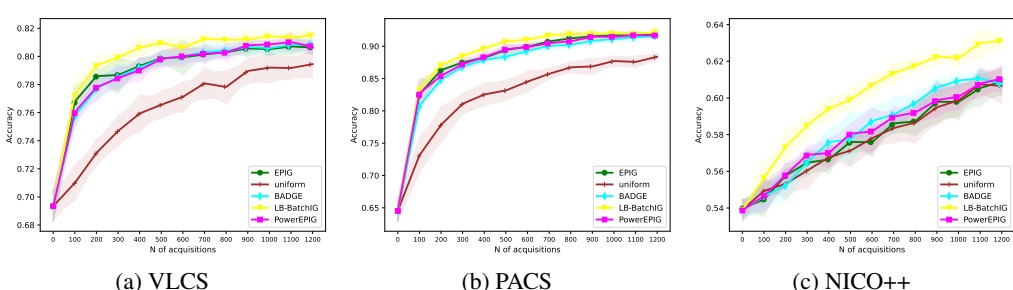

(a) VLCS  (b) PACS  (c) NICO++

Figure 7: The results on the image datasets, including VLCS, PACS and NICO++ benchmarks.

typical benchmarks of domain generalization in computer vision field to examine the effectiveness of our method.

**Experimental setup** We leverage three representative benchmarks respectively to construct datasets for experiments, that are VLCS (Fang et al., 2013), PACS (Li et al., 2017) and NICO++ (Zhang et al., 2023). The VLCS benchmark is composed by different data sources including PASCAL VOC (Everingham et al., 2010), Caltech101 (Griffin et al., 2007), LabelMe (Russell et al., 2008) and SUN09 (Choi et al., 2010) datasets. The PACS and NICO++ benchmarks consist of images from different domains. We control the proportion of different data sources or domains to manifest distribution shift among training dataset, candidate pool and target population. The detailed information about the composition of the training dataset, candidate pool and target population can be found in the Appendix C.

For the model architecutre, we adopt the Resnet18 (He et al., 2016) as the backbone and place dropout layer the last hidden layer with the dropout rate as 0.5.

**Result** We follow the same experiment settings as the previous experiments. Specifically, the batch size of single acquisition is set as $c = 50$, and we conduct the repeated experiments for 10 times.

The results are shown in Figure 7. The results further reinforce the conclusion we obtain in the previous experiments. Generally, the Uniform method underperforms the other methods since it ignores the distinction among the samples in improving model prediction and does not prefer the samples contributing more to the model performance. The BADGE and EPIG methods improve upon Uniform, but achieve suboptimal performance because of the distribution shift and redundant information problem respectively. PowerEPIG incorporate the diversity into acquisition process in a straightforward way. Our proposed acquisition function LB-BatchIG characterize the acquisition criteria at batch-level, which simultaneously considers the predictive information gain on the shifted target population and the diversity of acquired sample batch, and consistently achieves the best performance.

## 5 CONCLUSION AND LIMITATION

In this paper, we investigate how to acquire new samples from the candidate data pool for improving the model performance on a shifted target population. We propose a novel acquisition function LB-BatchIG that is built upon predictive information gain on the target population to address the distribution shift, while considering the diversity of acquired batch to alleviate the redundant information problem. We also utilize the submodular property of the acquisition function to solve the optimization problem with greedy algorithm. Extensive experiments on the different datasets demonstrate the effectiveness of our method.

The greedy algorithm is an approximation solution to the batch acquisition. Therefore, proposing a more effective optimization algorithm is worthy to research for future work. Besides, the method is limited to classification task. The extension to more complex tasks, such as object detection, is also a valuable research problem in the future.

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

APPENDIX

## A   LARGE LANGUAGE MODEL USAGE

In this paper, we claim that large language models (LLMs) are used solely to support and refinethe writing process. Specifically, we use LLMs to provide word-level and sentence-level suggestions to enhancethe overall fluency of the text.

## B   PROOF

**Theorem B.1.** *(Restated) According to the information theory, we have*

$$\text{LB-BatchIG}(\{\mathbf{x}_i^b\}) \leq \mathbb{E}_{p^t(\mathbf{x}^*)}\left[\text{BIG}(\{\mathbf{x}_i^b\}, \mathbf{x}^*)\right].$$

*Proof.* For arbitrary $i \in [c]$, we can prove that $\text{IG}(\mathbf{x}_i^b, \mathbf{x}^*) \leq \text{BIG}(\{\mathbf{x}_i^b\}, \mathbf{x}^*)$. Without loss of the generality, we hypothesize $i = 1$. This is equivalent to

$$\int_{y_1^b, y_2^b, \ldots, y_c^b, y^*} p(y_1^b, y_2^b, \ldots, y_c^b, y^* | \mathbf{x}_1^b, \mathbf{x}_2^b, \ldots, \mathbf{x}_c^b, \mathbf{x}^*; \mathcal{D}) \cdot$$

$$\log \frac{p(y_1^b, y_2^b, \ldots, y_c^b, y^* | \mathbf{x}_1^b, \mathbf{x}_2^b, \ldots, \mathbf{x}_c^b, \mathbf{x}^*; \mathcal{D})}{p(y_1^b, y_2^b, \ldots, y_c^b | \mathbf{x}_1^b, \mathbf{x}_2^b, \ldots, \mathbf{x}_c^b; \mathcal{D}) p(y^* | \mathbf{x}^*; \mathcal{D})} dy_1^b \ldots dy_c^b dy^*$$

$$\geq \int_{y_1^b, y^*} p(y_1^b, y^* | \mathbf{x}_1^b, \mathbf{x}^*; \mathcal{D}) \log \frac{p(y_1^b, y^* | \mathbf{x}_1^b, \mathbf{x}^*; \mathcal{D})}{p(y_1^b | \mathbf{x}_1^b; \mathcal{D}) p(y^* | \mathbf{x}^*; \mathcal{D})} dy_1^b dy^*$$

$$= \int_{y_1^b, y_2^b, \ldots, y_c^b, y^*} p(y_1^b, y_2^b, \ldots, y_c^b, y^* | \mathbf{x}_1^b, \mathbf{x}_2^b, \ldots, \mathbf{x}_c^b, \mathbf{x}^*; \mathcal{D}) \cdot$$

$$\log \frac{p(y_1^b, y^* | \mathbf{x}_1^b, \mathbf{x}^*; \mathcal{D})}{p(y_1^b | \mathbf{x}_1^b; \mathcal{D}) p(y^* | \mathbf{x}^*; \mathcal{D})} dy_1^b dy^*. \tag{12}$$

By move the l.h.s to the right side of inequality, it becomes

$$\int_{y_1^b} p(y_1^b | \mathbf{x}_1^b; \mathcal{D}) dy_1^b \int_{y_2^b \ldots, y_c^b, y^*} p(y_2^b, \ldots, y_c^b, y^* | \mathbf{x}_1^b, y_1^b, \mathbf{x}_2^b, \ldots, \mathbf{x}_c^b, \mathbf{x}^*; \mathcal{D}) \cdot$$

$$\log \frac{p(y_2^b, \ldots, y_c^b | \mathbf{x}_1^b, y_1^b, \mathbf{x}_2^b, \ldots, \mathbf{x}_c^b; \mathcal{D}) p(y^* | \mathbf{x}_1^b, y_1^b, \mathbf{x}^*; \mathcal{D})}{p(y_2^b, \ldots, y_c^b, y^* | \mathbf{x}_1^b, y_1^b, \mathbf{x}_2^b, \ldots, \mathbf{x}_c^b, \mathbf{x}^*; \mathcal{D})} dy_2^b \ldots dy_c^b dy^* \leq 0. \tag{13}$$

Because the function $\log(\cdot)$ is convex function, according to Jensen inequality, we can prove

$$\int_{y_2^b \ldots, y_c^b, y^*} p(y_2^b, \ldots, y_c^b, y^* | \mathbf{x}_1^b, y_1^b, \mathbf{x}_2^b, \ldots, \mathbf{x}_c^b, \mathbf{x}^*; \mathcal{D})$$

$$\log \frac{p(y_2^b, \ldots, y_c^b | \mathbf{x}_1^b, y_1^b, \mathbf{x}_2^b, \ldots, \mathbf{x}_c^b; \mathcal{D}) p(y^* | \mathbf{x}_1^b, y_1^b, \mathbf{x}^*; \mathcal{D})}{p(y_2^b, \ldots, y_c^b, y^* | \mathbf{x}_1^b, y_1^b, \mathbf{x}_2^b, \ldots, \mathbf{x}_c^b, \mathbf{x}^*; \mathcal{D})} dy_2^b \ldots dy_c^b dy^*$$

$$\leq \log \int_{y_2^b \ldots, y_c^b, y^*} p(y_2^b, \ldots, y_c^b, y^* | \mathbf{x}_1^b, y_1^b, \mathbf{x}_2^b, \ldots, \mathbf{x}_c^b, \mathbf{x}^*; \mathcal{D})$$

$$\frac{p(y_2^b, \ldots, y_c^b | \mathbf{x}_1^b, y_1^b, \mathbf{x}_2^b, \ldots, \mathbf{x}_c^b; \mathcal{D}) p(y^* | \mathbf{x}_1^b, y_1^b, \mathbf{x}^*; \mathcal{D})}{p(y_2^b, \ldots, y_c^b, y^* | \mathbf{x}_1^b, y_1^b, \mathbf{x}_2^b, \ldots, \mathbf{x}_c^b, \mathbf{x}^*; \mathcal{D})} dy_2^b \ldots dy_c^b dy^*$$

$$= \log \int_{y_2^b \ldots, y_c^b, y^*} p(y_2^b, \ldots, y_c^b | \mathbf{x}_1^b, y_1^b, \mathbf{x}_2^b, \ldots, \mathbf{x}_c^b; \mathcal{D})$$

$$p(y^* | \mathbf{x}_1^b, y_1^b, \mathbf{x}^*; \mathcal{D}) dy_2^b \ldots dy_c^b dy^*$$

$$= \log 1 = 0$$

Therefore, the Equation 13 is satisfied. By analog, we can prove $\text{IG}(\mathbf{x}_i^b, \mathbf{x}^*) \leq \text{BIG}(\{\mathbf{x}_i^b\}, \mathbf{x}^*), \forall 1 \leq i \leq c$.

We can derive that

$$\max_{1 \leq i \leq c} \text{IG}(\mathbf{x}_i^b, \mathbf{x}^*) \leq \text{BIG}(\{\mathbf{x}_i^b\}, \mathbf{x}^*)$$

$$\Rightarrow \text{LB-BatchIG}(\{\mathbf{x}_i^b\}) = \mathbb{E}_{p^t(\mathbf{x}^*)}\left[\max_{1 \leq i \leq c} \text{IG}(\mathbf{x}_i^b, \mathbf{x}^*)\right]$$

$$\leq \mathbb{E}_{p^t(\mathbf{x}^*)}\left[\text{BIG}(\{\mathbf{x}_i^b\}, \mathbf{x}^*)\right]$$

$\square$

**Proposition B.1.** *(Restated) Regarding the unlabeled pool as the element set $V$ in Definition 3.1 and the sample batch as the subset, the acquisition function LB-BatchIG is a normalized monotone non-decreasing submodular function.*

*Proof.* We respectively proof the property of normalization, monotone non-decreasing, and submodular.

**Normalization**: According to Equation 5, we can easily obtain that $\text{LB-BatchIG}(\varnothing) = 0$. Hence the normalization property is satisfied.

**Monotone non-decreasing**: For a batch $\{\mathbf{x}_i^b\}_{1 \leq i \leq d}$ and a sample $\mathbf{x}' \notin \{\mathbf{x}_i^b\}_{1 \leq i \leq d}$, we have

$$\text{LB-BatchIG}(\{\mathbf{x}_i^b\}_{1 \leq i \leq d} \cup \{\mathbf{x}'\})$$

$$= \mathbb{E}_{p^t(\mathbf{x}^*)}\left[\max\{\max_{1 \leq i \leq d} \text{IG}(\mathbf{x}_i^b, \mathbf{x}^*), \text{IG}(\mathbf{x}', \mathbf{x}^*)\}\right]$$

$$\geq \mathbb{E}_{p^t(\mathbf{x}^*)}\left[\max_{1 \leq i \leq d} \text{IG}(\mathbf{x}_i^b, \mathbf{x}^*)\right] = \text{LB-BatchIG}(\{\mathbf{x}_i^b\}_{1 \leq i \leq d}) \tag{14}$$

**Submodular**: For two arbitrary sample batches $A$ and $B$ and sample $\mathbf{x}^*$, we denote

$$a = \max_{\mathbf{x} \in A} \text{IG}(\mathbf{x}, \mathbf{x}^*), \tag{15}$$

$$b = \max_{\mathbf{x} \in B} \text{IG}(\mathbf{x}, \mathbf{x}^*), \tag{16}$$

$$c = \max_{\mathbf{x} \in A \cap B} \text{IG}(\mathbf{x}, \mathbf{x}^*). \tag{17}$$

Then we have $\max\{a, b\} = \max_{\mathbf{x} \in A \cup B} \text{IG}(\mathbf{x}, \mathbf{x}^*)$ and $a \geq c, b \geq c$.

If $a \geq b$, then $a = \max\{a, b\} \cap b \geq c \Rightarrow a + b \geq \max\{a, b\} + c$.

If $a < b$, then $b = \max\{a, b\} \cap a \geq c \Rightarrow a + b \geq \max\{a, b\} + c$.

Therefore, we have

$$\mathbb{E}_{p^t(\mathbf{x}^*)}[\max_{\mathbf{x} \in A} \text{IG}(\mathbf{x}, \mathbf{x}^*) + \max_{\mathbf{x} \in B} \text{IG}(\mathbf{x}, \mathbf{x}^*)] \tag{18}$$

$$\geq \mathbb{E}_{p^t(\mathbf{x}^*)}[\max_{\mathbf{x} \in A \cap B} \text{IG}(\mathbf{x}, \mathbf{x}^*) + \max_{\mathbf{x} \in A \cup B} \text{IG}(\mathbf{x}, \mathbf{x}^*)].$$

Based on this, we conclude

$$\text{LB-BatchIG}(A) + \text{LB-BatchIG}(B)$$

$$\geq \text{LB-BatchIG}(A \cap B) + \text{LB-BatchIG}(A \cup B)$$

$\square$

## C   DATA COMPOSITION OF EXPERIMENTS

The data composition in the experiments of synthetic data, tabular data and image data are as follows:

**Synthetic Data**

- Training dataset: Subpop-0 1200 + Subpop-1 20 + Subpop-2 180
- Candidate pool: Subpop-0 6000 + Subpop-1 $\alpha$ + Subpop-2 3400
- Target population: Subpop-1 3000

**Tabular Data**

Income Prediction

- Training dataset: CA 1000 + AL 100
- Candidate pool: CA 10000 + AL 10000 + PR 5000
- Target population: PR 2000

Employment Prediction

- Training dataset: CA 400 + AL 200
- Candidate pool: CA 10000 + AL 10000 + PR 5000
- Target population: PR 3000

**Image Data**

VLCS benchmark

- Training dataset: LABELME 400+CALTECH 400+ SUN 400
- Candidate pool: LABELME 1000 + CALTECH 400+ PASCAL 1000+SUN 1000
- Target population: PASCAL 1000

PACS benchmark

- Training dataset: photo 400+ art 400+sketch 400
- Candidate pool: photo 1000+ art 1000 cartoon 1000+ sketch 1000
- Target population: cartoon 1000

Nico++ benchmark

- Training dataset: autumn 400 +rock 400+dim 400 +grass 400
- Candidate pool: autumn 1000 + rock 1000 + dim 1000 + grass 1000 + outdoor 1000 + water 1000
- Target population: outdoor 1000 + water 1000

## D   PSEUDO-CODE OF OUR ALGORITHM

The pseudo-code of our algorithm is presented as Algorithm 1.

## E   RELATED WORK

In this section, we briefly review the related research of the unsupervised domain adaptation, active learning and active DA.

---

**Algorithm 1** Greedy algorithm for optimizing LB-BatchIG

---

1: **Input:** Training dataset $\mathcal{D}_{tr} = \{(\mathbf{x}_i^{tr}, y_i^{tr})\}_{1 \leq i \leq n_{tr}}$, unlabeled pool $\mathcal{D}_{po} = \{\mathbf{x}_i^{po}\}_{1 \leq i \leq n_{po}}$, the test samples representing the test distribution $\mathcal{D}_{te} = \{\mathbf{x}_i^{te}\}_{1 \leq i \leq n_{te}}$ and the model $f_\theta$ with parameter $\theta$.
2: **Output:** the sample batch $\mathcal{B} = \{\mathbf{x}_i^b\}_{1 \leq i \leq c}$
3: Learning the model $f_\theta$ on the training dataset $\mathcal{D}_{tr}$ and sample a collection of parameters $\{\theta_l\}_{1 \leq l \leq m}$ from posterior distribution $p(\theta|\mathcal{D}_{tr})$.
4: Initialize the sample batch $\mathcal{B}_0 = \varnothing$.
5: **for** $i = 1$ **to** $c$ **do**
6:     Sample a batch of test samples $\{x_j^*\}_{1 \leq j \leq s}$
7:     $b_i \leftarrow 1$
8:     $r \leftarrow 0$
9:     **for** $w = 2$ **to** $n_{po}$ **do**
10:         **if** LB-BatchIG$(\mathcal{B}_{i-1} \cup \mathbf{x}_w^{po}) > r$ **then**
11:             $b_i \leftarrow w$
12:             $r \leftarrow$ LB-BatchIG$(\mathcal{B}_{i-1} \cup \mathbf{x}_w^{po})$
13:         **end if**
14:         $\mathcal{B}_i = \mathcal{B}_{i-1} \cup \{\mathbf{x}_{b_i}^{po}\}$
15:     **end for**
16: **end for**
17: **return** the acquisition batch $\mathcal{B}_c$

---

### E.1 UNSUPERVISED DOMAIN ADAPTATION

When the target domain for generalization is known, a bunch of domain adaptation (DA) (Ben-David et al., 2010) works can be proposed. To match the feature distribution of source domains and target domains, some approaches re-weight or select the training samples (Jiang & Zhai, 2007; Huang et al., 2006). Besides, learning a feature transformation (Ganin & Lempitsky, 2015; Bousmalis et al., 2016; Tzeng et al., 2014) is an alternative method to align the feature distributions. Specifically, (Tzeng et al., 2014) leverage Maximum Mean Discrepancy (MMD) which characterizes the difference of distribution mean in reproducing kernel Hilbert space. (Ganin & Lempitsky, 2015) and (Ganin et al., 2016) train a domain classifier and applied the separability between domains as the discrepancy measurement. And some literature (Li et al., 2020; Courty et al., 2016) use transport distance to learn the domain-aligned transformation. Although noteworthy advancements have been made from the perspective of algorithms to enhance model performance on target domains, they still fall behind the supervised learning counterpart (Chen et al., 2018; Tsai et al., 2018). Therefore, it can play a significant role to enlarge the training dataset with beneficial samples for training the models.

### E.2 ACTIVE LEARNING

Active learning investigates how to acquire data for annotation to optimize the model. The proposed data acquisition criterions cover many aspects (Liu et al., 2022), including uncertainty, model influence, representativeness. For criterions based on uncertainty, (Joshi et al., 2009) calculate the prediction uncertainty by the entropy of classification probability and the difference of the highest two probability, (Ržička et al., 2020) take the gap between the highest probability and 1.0 as the uncertainty. For criterion based on representativeness, the samples central to the data distribution are acquired. (Settles & Craven, 2008) calculate the similarity to other samples as the representativeness metric. Core-set methods (Qin et al., 2021; Sener & Savarese, 2017a) try to choose center samples so that the largest distance between samples and the nearest center is minimized. As for criterions based on model influence, the methods select the samples having great impact on the model parameters if incorporated into training dataset. Some active learning algorithms (Sourati et al., 2017; 2019) apply Fisher information (Fisher, 1922) as the measurement of the impact on model parameters. Although Fisher information is theoretically grounded, it is computationally intensive in practice. BADGE (Ash et al., 2020) use the magnitude of gradient constituting the metric of impact on model parameters. Inspired by the bayesian inference (Bernardo & Smith, 2009), bayesian active learning (Gal et al., 2017; Kirsch et al., 2019; Smith et al., 2023) make assumptions on the

prior distribution of model parameters, and calculate the entropy reduction after adding samples as the model influence metric.

### E.3 ACTIVE DOMAIN ADAPTATION

Active DA focuses on acquiring samples from the target domain to accomplish domain adaptation. To be concrete, AADA (Su et al., 2020) selects samples based on uncertainty and domainness measured by a domain discriminator. (Fu et al., 2021) propose a unified criterion incorporating transferable committee, transferable uncertainty, and transferable domainness. CLUE (Prabhu et al., 2021) design a clustering algorithm weighted by uncertainty to select samples from target domain. DiaNA (Huang et al., 2023) propose a Divide-And-Adapt protocol which partitions the target samples into four types and selects the uncertain and inconsistent ones. These methods hypothesize a candidate pool with the same distribution to target domains can be acquired for annotation. However, the commercial restriction, privacy concerns and other issues can make this hypothesis unrealistic.

## F EXPERIMENTAL COMPUTE RESOURCE

All experiments are conducted with the following settings:

- Operating Systems: Ubuntu 14.04.1 LTS
- CPU: Intel(R) Xeon(R) CPU E5-2630 v2 @ 2.60GHz
- GPU: Nvidia RTX 3090 $\times$ 1
- Memory: 256GB

