# OpenReview forum: "Scalable Bayesian Active Learning with Batch Acquisition under Distribution Shift"
_ICLR.cc/2026/Conference — Submitted to ICLR 2026_

### Official Review · Reviewer_tCuD · 2025-10-26

**Soundness:** 3
**Presentation:** 3
**Contribution:** 3
**Rating:** 6
**Confidence:** 4

**Summary:**

This paper addresses the critical problem of performance degradation in machine learning models when faced with a distribution shift between training and testing data. The authors proposes a data-centric solution, focusing on active learning to acquire a batch of informative samples from a candidate pool to improve model generalization on a shifted target distribution. The work identifies key limitations in existing approaches: conventional active learning often assumes i.i.d. data, while Active Domain Adaptation (ADA) makes the restrictive assumption that the candidate pool is drawn from the target domain.

To overcome these challenges, the authors introduce a novel acquisition function, the "Lower Bound of Batch Information Gain" (LB-BatchIG). This function is designed to approximate the true, but intractable, information gain that a batch of samples provides for predictions on the target distribution. The paper's core theoretical contributions are twofold. First, it proves that LB-BatchIG is a valid lower bound on the true batch information gain. Second, and more significantly, it proves that LB-BatchIG is a submodular function. This property is crucial as it guarantees that an efficient greedy algorithm can find a near-optimal batch with an approximation factor of $(1 - 1/e)$ relative to the optimal solution.

The proposed method is empirically validated across a diverse set of experiments, including synthetic data, tabular data from US Census records, and standard image classification benchmarks for domain generalization (VLCS, PACS, NICO++). The results consistently show that the LB-BatchIG-based greedy selection outperforms several baselines, including Uniform sampling, a top-K version of EPIG, PowerEPIG, and BADGE, in improving model accuracy on the target distribution.

**Strengths:**

**Significance and Novelty of the Problem Formulation**

The paper tackles a highly relevant and practical problem in machine learning. The setting—where the training data, the unlabeled candidate pool, and the target data may all come from different distributions—is a realistic reflection of many real-world deployment scenarios where i.i.d. assumptions do not hold. By moving beyond the standard active learning paradigm and the restrictive assumptions of ADA, the work addresses a significant gap in the literature.

**Theoretical Soundness and Elegance**

The theoretical foundation of the proposed method is a major strength. The formulation of the batch acquisition problem as the maximization of a submodular function is elegant and provides strong formal justification for the proposed algorithm. Proving that the LB-BatchIG function satisfies the properties of submodularity is a non-trivial result that bridges the information-theoretic objective with a computationally tractable, near-optimal solution strategy.

**Weaknesses:**

**Insufficient Comparison with State-of-the-Art Baselines**

The most significant weakness of this work is the limited scope of its baseline comparisons. While the included methods (Uniform, EPIG, PowerEPIG, BADGE) are reasonable, they do not represent the full landscape of modern batch active learning, leading to potentially unsubstantiated claims of superiority. Several critical and more recent baselines are missing: BatchBALD, BAIT, SIMILAR, Cluster-Margin, etc.

**Insufficient Analysis of Practical Scalability**

**Lack of Empirical Time Comparison:** For a paper claiming scalability as a key contribution, the lack of any empirical analysis of runtime is a significant omission. There are no wall-clock time comparisons against the baselines (e.g., BADGE, which is likely much faster) nor any analysis of how the runtime scales with key parameters like pool size $n_{po}$ and batch size $c$.

**Lack of Space Complexity Analysis:** The paper provides no discussion of the memory footprint (space complexity) of the proposed method. A comprehensive scalability analysis should compare both time and space complexity against other published models to truly substantiate the "Scalable" claim.

**Reliance on MC Dropout for Bayesian Approximation**

The performance of the proposed method is fundamentally coupled with the quality of the posterior approximation provided by MC Dropout. While MC Dropout is a practical and widely used technique, it is known to have limitations and may not always produce well-calibrated or reliable uncertainty estimates, especially for the out-of-distribution data that is central to this paper's problem setting. The paper does not explore the sensitivity of LB-BatchIG to the choice of Bayesian approximation. It is an unstated assumption that the uncertainty estimates are meaningful enough to guide acquisition for OOD generalization. An analysis with more sophisticated or robust methods for uncertainty quantification (e.g., deep ensembles, SGLD, SGHMC, cSG-MCMC) would strengthen the paper's claims of general utility.

**Absence of a Dedicated Related Work Section**

The discussion of related work is relegated to an appendix (Appendix E), which diminishes its visibility and integration with the main body of the paper. A dedicated "Related Work" section within the main text is standard practice and essential for properly contextualizing the paper's contributions. This section should clearly position the proposed method with respect to prior work in Bayesian active learning, batch-mode active learning, and active learning for domain adaptation, highlighting the specific gaps the paper aims to fill. Its current placement makes it difficult for readers to appreciate the novelty and significance of the work without cross-referencing the appendix.

**Questions:**

1. Regarding Baselines: Could the authors comment on the omission of BatchBALD, SIMILAR, BAIT etc as a baseline? Given its prominence and direct relevance, a comparison seems essential for substantiating the paper's performance claims. What are the expected advantages of the proposed prediction-oriented, target-aware formulation over BatchBALD's parameter-oriented approach in the context of distribution shift?

2. Regarding Scalability: The title emphasizes scalability, yet the paper lacks an empirical analysis of computational cost. Could you provide wall-clock time comparisons against the baselines, particularly the likely faster BADGE method? Furthermore, how does the runtime of the greedy selection algorithm scale in practice with the pool size $n_{po}$ and batch size $c$?

3. Regarding Hyperparameters: The estimation of LB-BatchIG relies on the number of posterior samples ($m$) and target samples ($s$). How were these values selected for the experiments? Was a sensitivity analysis performed to understand the trade-off between the quality of the acquisition function's estimate and the associated computational cost?

---

### Official Review · Reviewer_NzJm · 2025-10-30

**Soundness:** 2
**Presentation:** 3
**Contribution:** 2
**Rating:** 4
**Confidence:** 4

**Summary:**

This paper proposes a batch acquisition method for Bayesian active learning under the distribution shift scenario. It designs a new acquisition function LB-BatchIG as a lower bound of BIG. Then, it provides an algorithm based on the sub-modular property of the objective function. The paper also conducts experiments on synthetic data, tabular data, and image data to show the effectiveness.

**Strengths:**

1. The distribution shift problem in active learning is interesting.
2. The method is reasonable. The paper also provides a theoretical analysis of the acquisition function.
3. The experimental results are good.

**Weaknesses:**

1. According to the Introduction, this paper seems to be an improvement of EPIG [1], where EPIG selects the sample with the highest criteria, whereas this paper tries to select a batch of samples. The improvement approach in this paper to EPIG is a little similar to the improvement approach in the Paper "BatchBALD: Efficient and Diverse Batch Acquisition for Deep Bayesian Active Learning" [2] to Paper "Bayesian active learning for classification and preference learning" [3].

[1] Prediction-oriented bayesian active learning, in International Conference on Artificial Intelligence and Statistics, 2023.
[2] BatchBALD: Efficient and Diverse Batch Acquisition for Deep Bayesian Active Learning, in NeurIPS 2019.
[3] Bayesian active learning for classification and preference learning, arXiv preprint arXiv:1112.5745, 2011.

2. The paper highlights "scalable", but the experiments do not evaluate whether it is scalable. The experiments should be conducted on large-scale data sets. The comparison of running time should also be reported in the experiments.

3. The compared methods are insufficient. Besides Uniform, only 3 methods are compared, and the newest method is EPIG, which is proposed in 2023. More current methods should be compared.

**Questions:**

Please see Weaknesses.

---

### Official Review · Reviewer_kHnb · 2025-10-30

**Soundness:** 2
**Presentation:** 1
**Contribution:** 3
**Rating:** 2
**Confidence:** 3

**Summary:**

The paper introduces a new Bayesian active learning strategy called LB-BatchIG, designed to select batches that improve generalization under distribution shift. The authors propose an acquisition criterion, demonstrate that it is submodular, and employ a greedy selection algorithm for efficient batch construction. Experiments conducted on both synthetic and real-world datasets demonstrate that their greedy selection approach yields strong performance compared to baseline strategies.

**Strengths:**

- The authors address an important and underexplored problem of active learning under distribution shift with batch acquisition from diverse candidate pools.
- The authors provide theoretical guarantees, i.e., prove that the proposed acquisition function is submodular, enabling efficient greedy optimization.
- The strategy yields strong performance across diverse datasets, including synthetic, tabular, and vision benchmarks.

**Weaknesses:**

- **Clarity and Language:** The manuscript would benefit from substantial language editing, as grammatical errors throughout the text hinder readability and make it challenging to fully understand the authors' intended arguments. Additionally, several passages lack precision and could be explained more clearly. Improving language quality and providing more precise explanations would significantly enhance the manuscript's overall clarity and impact.
- **Comparison with EPIG:** The proposed LB-BatchIG criterion appears highly similar to the existing EPIG method, as it essentially takes the maximum of information gain, which corresponds to the inner expectation of the criterion in Eq. 2. This similarity raises important questions that should be addressed: What is the theoretical relationship between these objectives? Under what conditions does LB-BatchIG provide advantages over EPIG? A thorough theoretical comparison and discussion of these closely related approaches is necessary to justify the contribution of the proposed method.
- **Limited Baselines:** The experimental comparison is too limited, as it includes only EPIG, PowerEPIG, and BADGE as baselines. Several recent and effective active learning strategies are missing, such as BAIT [1], Typiclust [2], and UHerding [3], all of which have demonstrated strong performance in recent literature. Comparisons against these state-of-the-art methods are essential to properly evaluate the proposed approach and establish its contribution relative to current best practices in the field.
- **Missing Computational Analysis:** The evaluation lacks empirical validation of computational efficiency. While theoretical complexity is considered, no runtime experiments have been conducted to compare LB-BatchIG against baselines. To assess practical applicability, the authors should present query time comparisons across varying pool and batch sizes.

[1] Ash, Jordan, et al. "Gone fishing: Neural active learning with fisher embeddings." *Neurips, 2021.*

[2] Hacohen, Guy, Avihu Dekel, and Daphna Weinshall. "Active Learning on a Budget: Opposite Strategies Suit High and Low Budgets." *ICLR*, 2022.

[3] Bae, Wonho, Danica J. Sutherland, and Gabriel L. Oliveira. "Uncertainty Herding: One Active Learning Method for All Label Budgets." *ICLR*. 2025.

**Questions:**

I am open to revising my assessment based on the authors' responses to the weaknesses identified above, especially if additional experiments or clarifications can be provided.

---

### Official Review · Reviewer_Ctua · 2025-11-01

**Soundness:** 3
**Presentation:** 2
**Contribution:** 2
**Rating:** 4
**Confidence:** 4

**Summary:**

This paper introduces a batch active learning method designed for situations where the data distribution shifts between training and deployment. Building on expected predictive information gain, this paper proposes a lower-bound batch objective that encourages selecting diverse and informative examples aligned with the target distribution. This objective is submodular, allowing an efficient greedy selection strategy with theoretical guarantees.

**Strengths:**

The main idea of this paper is interesting, they use a lower-bound formulation of batch predictive information gain with submodularity properties. This is a creative adaptation of existing information-theoretic principles that enables practical and theoretically grounded batch acquisition. The method is clearly described, and the motivation for reducing redundancy and aligning selection with the target distribution is also meaningful.

**Weaknesses:**

1. Novelty issue: The method feels like a practical extension of existing information-gain based active learning rather than a clearly new acquisition paradigm.
2. The paper motivates the approach to improve the myopic selection in single-point information gain. However, the method itself uses a greedy selection procedure over a lower-bound surrogate.
3. Batch active learning inherently involves interactions among selected points. But this current formulation assumes that the maximal single-point contribution approximates joint utility.

**Questions:**

1. Why did this proposed method not compare with direct baselines such as BatchBALD and BAIT, which are highly relevant?
2. The experiments mainly show covariate shift. Have the authors considered other shift settings?
3. Can the authors report the runtime table to support the scalability claim?

---

### Meta-Review · Area_Chair_W6WW · 2026-01-13

**Summary:**

This paper addresses batch Bayesian active learning under distribution shift, where the training data, unlabeled candidate pool, and target deployment distribution may differ. It proposes a new acquisition function, LB-BatchIG, defined as a tractable lower bound on batch predictive information gain with respect to the target distribution, and proves that this objective is submodular. Leveraging submodularity, the paper uses a greedy algorithm with approximation guarantees to efficiently select informative and diverse batches. Experiments on synthetic, tabular, and image datasets under distribution shift show improved target-domain performance compared to several baseline active learning methods.

**Reviewer Concerns:**

Reviewers expressed concerns that the paper’s novelty is limited, viewing the proposed method as a natural batch extension of existing information-gain–based Bayesian active learning approaches such as EPIG and BatchBALD rather than a fundamentally new acquisition paradigm. They also noted that key state-of-the-art baselines were missing and that the paper’s central claim of scalability was not supported by empirical runtime or large-scale evaluations. Additionally, reviewers raised issues about clarity and writing quality, limited coverage of distribution shift settings, and the lack of analysis on sensitivity to the chosen Bayesian uncertainty approximation.

**Reviewer Scores:**

All reviewers are leaning towards rejection and there was no rebuttal.

---

### Decision · Program_Chairs · 2026-01-26

Reject